

# Downwind evolution of the volatility and mixing state of near-road aerosols near a US interstate highway

Provat Kumar Saha[1], Andrey Khlystov[2], and Andrew Patrick Grieshop[1]

[1]Department of Civil, Construction and Environmental Engineering, North Carolina State University, Raleigh, North Carolina, USA

[2]Division of Atmospheric Sciences, Desert Research Institute, Reno, Nevada, USA

*Correspondence to*: A. P. Grieshop (apgriesh@ncsu.edu)

**Abstract.** We present spatial measurements of particle volatility and mixing state at a site near a North Carolina interstate highway (I-40) applying several heating (thermodenuder; TD) experimental approaches. Measurements were conducted in summer 2015 and winter 2016 in a roadside trailer (10 m from road edge) and during downwind transects at different distances from the highway under favorable wind conditions using a mobile platform. Results show that the relative abundance of semi-volatile species (SVOCs) in ultrafine particles decreases with downwind distance, consistent with the dilution and mixing of traffic-sourced particles with background air and evaporation of semi-volatile species during downwind transport. An evaporation kinetics model was used to derive particle volatility distributions by fitting TD data. While the TD-derived distribution apportions about 20-30% of particle mass as semi-volatile (SVOCs; effective saturation concentration, $C* \geq 1\mu$ $m-3$) at 10 m from road edge, approximately 10% of particle mass is attributed to SVOCs at 220 m, showing that the particle-phase semi-volatile fraction decreases with downwind distance. The relative abundance of semi-volatile material in the particle-phase increased during winter. Downwind spatial gradients of the less-volatile particle fraction (that remaining after heating at 180°C) was strongly correlated with black carbon (BC). BC size distribution and mixing state measured using a Single Particle Soot Photometer (SP2) at the roadside trailer showed that a large fraction (70-80%) of BC particles were externally-mixed. Heating experiments with a volatility tandem differential mobility analyzer (V-TDMA) also showed that the non-volatile fraction in roadside aerosols are mostly externally mixed. V-TDMA measurements at different distances downwind from the highway indicate that mixing state of roadside aerosols does not change significantly (e.g., BC mostly remains externally mixed) within a few hundred meters from the highway. A preliminary analysis indicates that a super-position of volatility distributions measured in laboratory vehicle tests and of 'background' aerosol can be used to represent the observed partitioning of near-road particles. The results from this study highlight that exposures and impacts of BC and semi-volatile organics containing particles in a near-road microenvironment may differ across seasons and under changing ambient conditions.



1. Introduction

Motor vehicles are a large source of ambient fine particulate matter (PM) (Dallmann and Harley, 2010; Fraser et al., 1999; Kumar et al., 2011; Zhang et al., 2015). Particles emitted from vehicle exhaust are dominated by ultrafine particles (diameters < 100 nm) (Kleeman et al., 2000; Robert et al., 2007; Zhu et al., 2002), which are a concern due to their potential impacts on public health (Health Effects Institute, 2010; Hoek et al., 2009; Pope and Dockery, 2006). Vehicle-emitted PM largely consists of primary organic aerosol (POA) and black carbon (BC) (Dallmann et al., 2014; Maricq, 2007). Upon emission, vehicle exhaust undergoes rapid cooling and dilution with ambient air on the road. Emissions undergo further evolution from road to background-like conditions within a few hundred meters downwind from roadway (Robinson et al., 2010; Zhang et al., 2004), which involves complex physicochemical processes. Subsequently, size distribution and physio-chemical characteristics, and thus exposure characteristics and impacts, of aerosols evolve with downwind transport.

A large portion of POA emitted from motor vehicle is semivolatile material (Grieshop et al., 2009; May et al., 2013a; Presto et al., 2012) that can dynamically partition into the gas or particle phases with changing ambient conditions (e.g., temperature, concentrations) and atmospheric aging (Robinson et al., 2007, 2010). At equilibrium, the volatility of organic species (saturation vapor pressure, or equivalently, saturation concentration, $C^*$) dictates gas-particle partitioning (Donahue et al., 2006). Enthalpies of vaporization ($\Delta H_{vap}$) also influences the change in partitioning with temperature (Epstein et al., 2010; Ranjan et al., 2012). Depending on the volatility of POA, and atmospheric perturbations (dilution, changing temperature), semi-volatile species in POA will dynamically partition into gas or particle phases as they move downwind. Therefore, gas-particle partitioning of POA likely plays an important role in determining human exposure to traffic emitted particles under varied ambient conditions.

Fresh BC particles emitted from vehicles are typically fractal in morphology (Bond et al., 2013; China et al., 2014) and may have varying size, shape and mixing state. BC may exist in the same particle as OA and others species (internally mixed) or in separate particles (externally mixed). The mixing state and morphology of BC particles can influence their radiative absorption properties (Cappa et al., 2012) and deposition in the human respiratory tract (Broday and Rosenzweig, 2011). The mixing state and physio-chemical characteristics of BC particles evolve as they undergo atmospheric processing and aging (Adachi and Buseck, 2013; Subramanian et al., 2010). For example, photochemical oxidation of volatile organic compounds (VOCs), intermediate-VOCs (IVOCs), and semi-volatile organic compounds (SVOCs) form condensable vapors. These condensable vapor can partition into the particle-phase either by absorbing into the organic condensed phase or adsorbing onto nonvolatile BC cores (Donahue et al., 2006; Pankow, 1994; Roth et al., 2005).

Spatial measurements of volatility and mixing state of near-road aerosols are critically important to better understand the evolution of vehicle-emitted POA and BC under diverse ambient conditions (e.g., seasons) as they are transported from roadways, to assess human exposure and health risks, and to improve their representation in air quality and exposure

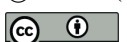



assessment models. Although a number of laboratory and field studies have investigated the volatility (Kuwayama et al., 2015; May et al., 2013a, 2013b, Li et al., 2016, Biswas et al., 2007; Grieshop et al., 2009) and mixing state (Tiitta et al., 2010, Liu et al., 2014, China et al., 2014, Willis et al., 2016) of traffic-emitted particles using various techniques, they have largely focused on measurements of sources or at fixed ambient location. To the best of our knowledge, no studies have been conducted

to systematically explore the evolution of volatility and mixing state of near-road aerosols at different distances from the roadway under diverse environmental conditions.

We measured the evolution of a highway plume at different downwind distances under diverse environmental conditions under favorable wind directions during summer and winter field campaigns. Heating (using a thermodenuder) experimental data coupled with a mass transfer kinetics model were used to investigate particle volatility, and heating (V-TDMA: Volatility

Tandem Differential Mobility Analyzer) and single particle data (SP2: Single Particle Soot Photometer) were used to explore mixing state of particles. The objectives of the study were to: (i) determine spatial distribution of aerosol volatility and mixing state in a near-road microenvironment, (ii) explore the influence of seasonality and ambient conditions on the phase-partitioning of near-road aerosols, and (iii) evaluate the representativeness of laboratory-derived POA volatility distributions from vehicle exhaust to explain real-world observations of aerosol volatility in a complex near-road microenvironment.

**2. Methods**

**2.1. Measurement sites**

Two month-long measurement campaigns were conducted at a site near Interstate 40 (I-40), outside Durham, North Carolina (35.865°N, 78.820°W) in Summer 2015 (June 1 –July 2) and Winter 2016 (January 18 – February 20). A map of the measurement site is shown in Figure S1. Detailed descriptions of the measurement site and campaigns are included in Saha et

al. (2017a); they are only described briefly here. At the measurement location, I-40 has eight lanes and an annual average daily traffic volume of 140 to 145 thousand vehicles per day (4-6% of which are heavy-duty diesel vehicles (HDDV)). At our site, I-40 adjoins a low-traffic rural road running almost perpendicular to the highway, in-line with the dominant wind direction (southwest; 225°). This minor road allows us to study the evolution of the highway plume at different downwind distances. Measurements were collected at a fixed site trailer 10 m from the highway (continuous) and during downwind transects on the

minor roadway at different downwind distances (10, 50, 100, 150, 220 m) from the highway (intermittent). Downwind transects measurements were performed on weekdays with the wind coming off of the highway consistently in summer (4 days) and winter (3 days). A mobile platform (van) was used for transects measurements. For a particular transect run, a sampling period at a downwind point was ~ 20 minutes; 4-5 downwind points were sampled consecutively. During sampling, the van engine was off to avoid self-contamination.  A 'full transect run' took approximately 2 hours. Typically, three transect runs were

completed per day: morning rush-hour (~7:00-9:00 am), mid-day (~12:30-2:30 pm) and evening rush hour (~4:30-6:30 pm).





To explore the spatial distribution of volatility and mixing state of particles, measurements collected during transect runs are the major focus of this paper.

## 2.2. Measurements

### 2.2.1 Thermodenuder (TD) experiments

Various configurations of heating (thermodenuder; TD) experimental approaches were applied to explore the volatility and mixing state of near-road aerosols. The methods applied here fall into two general categories: (i) heating of the polydisperse particle distribution (Huffman et al., 2008; Lee et al., 2010; Saha et al., 2017b), and (ii) heating of differential mobility analyzer- (DMA) selected monodisperse particles (volatility tandem DMA approach; V-TDMA) (Biswas et al., 2007; Kuhn et al., 2005; Tiitta et al., 2010). A custom-built, multi-tube thermodenuder (MT-TD) system was used for high time resolution

volatility measurements. The MT-TD consists of four separate heated lines controlled by automated valves that can be switched in approximately 1 second, enabling quick alternation between four different set temperatures. While measuring the evaporation of a polydisperse distribution, the MT-TD was coupled with a Scanning Mobility Particle Sizer (SMPS, TSI Inc., 3010 CPC, 3081 DMA; scan time 2.5 minutes) to measure thermodenuded distributions (10-400 nm) after heating at 60, 90, 120, and 180 ℃ with a residence time (Rt) of 30 s. All Rt values reported in this paper are volumetric residence time

(volume/flow rate) at room temperature, unless otherwise stated. Dump flows were used to maintain constant flow in all lines during MT-TD operation. A full set of temperature scans required ~ 10 minutes. Another SMPS (3010 CPC, 3081 DMA; scan time 2.5 minutes) continuously measured particle size distributions (10-400 nm) at ambient temperature. In some transect runs, the V-TDMA configuration was used. In this approach, DMA size-selected monodisperse particles (25, 50, 100, 250 nm) were heated at different TD temperatures (60-180°C) with a Rt of 30 s and the thermodenuded distributions measured using an

SMPS.

     In the stationary roadside trailer, a temperature stepping TD (TS-TD) (Huffman et al., 2008; Saha et al., 2017b) was continuously operated at 4 temperature steps (60, 90, 120, 180°C; Rt = 30 s) upstream of an Aerosol Chemical Speciation Monitor (ACSM, Aerodyne Inc.; 75-650 nm) and SMPS (10-400 nm). In this configuration, instruments were alternated between the bypass (ambient) and TS-TD lines at 10 minutes intervals using an automated 3-way valve. TD/ACSM provides

chemically resolved (organic, sulfate, nitrate, ammonium, and chloride) volatility data. Because TD data at different residence times provides additional constraints on the volatility parameter extraction process (Saha et al., 2015, 2017b), TD/SMPS data (10-400 nm) using the MT-TD setup were collected over a wide range of temperature and Rt conditions (T= 60, 90, 120 °C; Rt = 9, 13, 19, 30 s) during some of the summer campaign at the roadside trailer. An extra flow controller was used to vary Rt (Saha et al., 2015). In all measurements, a silica gel diffusion dryer was placed upstream of TD inlets and aerosol instruments

to maintain relative humidity (RH) < 30-40%.





### 2.2.2 Single Particle Soot Photometer (SP2)

A SP2 (Droplet Measurement Technology; DMT Inc.) was deployed at the roadside trailer during the winter campaign to measure the size distribution and mixing state of BC. The SP2 uses a laser-induced incandescence (Nd:YAG laser; 1064 nm) technique (Stephens et al., 2003) to measure refractory BC mass (rBC) in individual particles. The rBC containing particles passing through the laser beam scatter laser light and at the same time absorb energy and are heated to their vaporization temperature and incandesce (McMeeking et al., 2011a; Moteki and Kondo, 2007; Shiraiwa et al., 2007; Stephens et al., 2003). The incandescent light is proportional to rBC core mass. The SP2 incandescence response was calibrated with DMA-selected dried fullerene soot particles. A calibration curve is derived from the SP2 incandescence response and mass of the calibration particles from the mobility diameter and assuming an effective density of 1.8 g cm$^{-3}$. The scattering detectors were calibrated using dried PSL (polystyrene latex spheres) particles by relating the detector response to the PSL sizes. Ambient particles were dried before introduction into the SP2.

### 2.2.3 Other supporting measurements

Measurements of traffic (volume, composition, speed) and meteorological data (ambient T, RH, wind speed, and direction) and various gaseous and particulate air pollutant concentrations were collected throughout the campaigns, and are discussed in detail in Saha et al. (2017a). A 10-meter meteorological tower recorded meteorological data at the roadside trailer location. An existing remote traffic microwave sensor (RTMS) maintained by the North Carolina Department of Transportation (NC-DOT) provided traffic data. Particle size distributions (SMPS; 10-400 nm), chemical composition (ACSM; 75-650 nm), BC (Photo-acoustic Extinctiometer; PAX-870, Droplet Measurement Technology; DMT Inc.), NO/NO$_2$ (2B Technology 401/410), CO$_2$ (Li-cor Li-820) were continuously measured from the roadside trailer. During transect runs, particle sizing (SMPS), NO, NO$_2$, BC, and CO$_2$ instruments from the trailer were placed in the transect van to collect these parameters at different distances from the highway. Particle size distribution (SMPS; 10-400 nm), NO/NO$_2$ (Ecotech 9841), BC (microAeth, AE51) were continuously monitored in an upwind stationary background site, located on the opposite side of I-40, approximately 400 m away from the highway.

### 2.3 Data reduction

Evaporation of particles at a particular TD operating condition (T, Rt) is described in terms of volume fraction remaining (VFR; for SMPS data) or mass fraction remaining (MFR; for ACSM data). VFR (MFR) is the ratio of volume (mass) concentration measured in heated line (C$_{TD}$) to that in unheated bypass line (C$_{BP}$). The size distribution for the heated, size-selected monodisperse particles (V-TDMA approach) was typically bimodal; one mode did not shrink upon heating (non-volatile mode) and the other did (volatile mode). VFR for the size-selected particles was estimated as (D$_{p,heated}$)$^3$/(D$_{p,ambient}$)$^3$, where D$_{p,ambient}$ is the mode diameter of the selected monodisperse particles at ambient temperature and D$_{p,heated}$ is the mode diameter of the volatile mode after heating at a particular temperature. Therefore, estimated VFR for the size-selected particles



excludes the non-volatile population. Empirical particle loss correction factors (Saha et al., 2015), estimated as a function of TD operating conditions (T, Rt), were applied to the VFR from integrated SMPS volume and MFR from ACSM data. Because VFR for size-selected particles was calculated from the change in mode diameter, particle loss correction factors are not required in this calculation.

SP2 data were processed using the PSI (Paul Scherrer Institute) SP2 Toolkit. The rBC-containing particles are treated as an rBC core coated by a shell of other material. The size distribution of rBC-cores was derived based on the mass equivalent diameter (MED) of an rBC-core assuming a density of 1.8 cm$^{-3}$. The delay time between the peak of the incandescence and scattering signals is an indicator of the coating thickness (mixing state) (Moteki and Kondo, 2007), and was used to determine the number fraction of 'thinly coated' and 'thickly coated' rBC particles (McMeeking et al., 2011a; Shiraiwa et al., 2007;
Subramanian et al., 2010).

## 2.4 Parameterizing volatility

An evaporation mass transfer kinetics model (Lee et al., 2011) was applied to infer particle volatility distributions by fitting TD data. The volatility distribution extraction framework used here is similar to that described in Saha et al. (2015). The resulting fit empirically describes the particle volatility distribution using a volatility basis set (VBS) framework (Donahue et
al., 2006, 2012), where the material is lumped over a logarithmically spaced set of C* (effective saturation concentration) bins at a reference temperature of 25°C. A set of $f_i$ describes the distribution of semi-volatile species (particle + gas phase) in selected C* bins under a gas-particle equilibrium and is usually known as volatility distribution. A 6 bin $\log_{10}$VBS with C* bin range of $10^{-4}$ µg m$^{-3}$ to $10^1$ µg m$^{-3}$ at 25 °C was selected to describe the particle volatility distribution empirically. Before the TD inlet, an initial gas-particle equilibrium at ambient temperature (summer 30°C, winter 5 °C) and campaign-average
aerosol mass loading ($C_{OA}$ ~5µg m$^{-3}$) were assumed. The Clausius-Clapyeron equation was applied to calculate temperature dependent C* (Saha et al. 2015).

The mass transfer kinetics model tracks particle and gas-phase concentrations of the surrogate species (represented by C* bins) as they pass through the TD. The TD-derived volatility distributions from kinetics model fits are sensitive to assumptions of enthalpy of vaporization ($\Delta H_{vap}$) and evaporation coefficient ($\gamma_e$); these values are generally unknown *a priori* (Cappa and
Jimenez, 2010). $\gamma_e$ is often assumed to be in unity in fitting TD data (Grieshop et al., 2009; Li et al., 2016); however, recent studies reported a $\gamma_e$ values between 0.01 and 1 for different aerosol systems (Cappa and Jimenez, 2010; Saha et al., 2017b; Saha and Grieshop, 2016; Saleh et al., 2013). Similarly, in literature, different ranges of $\Delta H_{vap}$ values are reported for different aerosol systems (Epstein et al., 2010; May et al., 2013c; Ranjan et al., 2012). TD data collected at varying (T, Rt) provides additional constraints on feasible $\gamma_e$ and $\Delta H_{vap}$ values (Saha et al., 2015, 2017b; Saha and Grieshop, 2016). TD data over a
wide range in (T, Rt) space were collected during the I-40 summer campaign at the near-road trailer, and are shown in Figure S2 (a-d). Following Saha et al. (2015, 2017b), we used this data set to optimize a set of $\gamma_e$ and $\Delta H_{vap}$ values that best explain the evaporation observed in near-highway aerosols (see Figure S2e for details). A $\gamma_e$ of 0.25 and $\Delta H_{vap}$ of 100 kj mol$^{-1}$ provided



the overall best fit for this data set. We adopted these estimated $\gamma_e$ and $\Delta H_{vap}$ values for the near-highway aerosol system for further fitting of TD data from different distances from the highway across all seasons and sizes. Saha et al. (2017b) reported similar $\gamma_e$ and $\Delta H_{vap}$ values for ambient TD data from two sites in the southeastern US under diverse conditions. Saleh et al. (2012) derived a $\gamma_e$ value of 0.28-0.46 for ambient aerosols in Lebanon. Therefore, given the consistency in reported $\gamma_e$ and

$\Delta H_{vap}$ values across diverse settings (Saha et al., 2017b; Saleh et al., 2012),  it is reasonable to use constant values for further fitting of TD data from the same site under different conditions. Other inputs to the mass transfer model include diffusion coefficient (D), surface tension ($\sigma$), molecular weight (MW) and density ($\rho$); the assumed values generally have a smaller influence on modeled evaporation in TDs (Cappa and Jimenez, 2010; Saha et al., 2015), and are approximated from literature (Table S1).

**3. Results and discussion**

**3.1 Observed evaporations in TD with downwind distance**

Figure 1 shows the measured VFR at 60°C as a function of distance from the highway. The particle volume fraction that evaporates at low and moderate TD temperature (e.g., 1-VFR at 60˚C) consists of semi-volatile species, presumably OA. VFR data are shown for different monodisperse particle sizes (e.g., 25, 50, 100, 250 nm), and for the integrated volume of polydisperse distributions. Results shows that the evaporation observed in a TD at 60°C decreases with downwind distance

during transects in both seasons, which suggests a reduction in relative abundance of the semivolatile fraction in particles with distance. This reduction is especially pronounced over the ultrafine particle range (<100 nm). Two plausible reasons could contribute to this observation. First, a fraction of semi-volatile species in vehicle-emitted fresh particles may be evaporating during transport due to dilution-driven processes (Choi and Paulson, 2016; Robinson et al., 2007; Shrivastava et al., 2006).

Second, since the concentration of vehicle-emitted particles decreases rapidly with distance from the highway, the relative proportion of background particles in the sampled aerosol (vehicle-emitted + background) increases with distance. If one assumes that background particles are less volatile than vehicle-emitted fresh particles, the relative abundance of the less-volatile material in the sampled aerosols will increase with distance. The influence of each factor cannot be isolated directly from TD measurements. However, the particle size distributions measured at background (upwind) and downwind locations

from the highway (Figure S3) indicate that vehicle-emitted fresh particles are dominated by ultrafine particles (< 100 nm), while background particles are predominantly in a relatively larger mode. When polydisperse particles (10-400 nm) were heated at a moderate TD temperature (60°C), the changes in the larger size range (>100 nm) was observed to be minimal (Figure S4). Larger particles also do not show significant downwind gradients in evaporation upon heating at 60°C (Figure S4). Therefore, the observed downwind decrease in evaporation of ultrafine particles at 60°C is likely more influenced by the

dilution-driven losses of semi-volatile species during downwind transport.





The general trends in evaporation at 60°C observed as a function of downwind distance were consistent between summer and winter (Figure 1). However, the evaporation observed in winter was slightly higher than that in summer, specifically closer to the highway and for smaller particles. This observation is consistent with that of Kuhn et al. (2005), who reported greater evaporation of near-road particles in winter at a particular TD temperature. Two possible factors may contribute to this inter-

seasonal difference. First, the initial partitioning of SVOCs entering the TD is different; at colder temperatures, a higher fraction of semi-volatile materials is expected to partition into the particle-phase. An analysis of temperature effects on the partitioning of semi-volatile materials from vehicular emissions (see Figure S5) indicates that while 40-70% of semi-volatile emissions reside in the particle-phase under typical winter conditions (0-10°C), only 10-20% do so under summer conditions (20-30°C). This analysis used the gasoline POA volatility distribution from May et al. (2013a) and $\Delta H_{vap}$ from Ranjan et al.

(2012) and considered a range of OA concentrations for a typical roadside environment (e.g., 0.5 to 5 µg m$^{-3}$). Second, the difference could be due inter-seasonal differences in emission properties (the volatility of what is emitted) and atmospheric dilution. The effect of these two effects cannot be isolated directly from TD observations, but application of an evaporation kinetics model can disentangle them to some extent. For example, during modeling, initial gas-particle equilibrium at ambient temperatures (winter ~5 °C, summer ~30°C) was assumed before the TD inlet, which will account for the ambient temperature

effect on initial SVOC partitioning. Other effects (if any) should be reflected in the resulting fitted volatility distributions; modeling results are discussed in Sec. 3.3.

Figure 2 shows the evaporation observed at a higher TD temperature (180°C). VFR of PM$_{0.4}$ (integrated volume between 10-400 nm) at 180 °C decreases with downwind distance. The particle volume fraction that does not evaporate at 180 °C will consist of BC, other refractory materials (e.g. metals, crustal materials) and/or extremely low volatile organics (ELVOCs;

$C^* < 10^{-3}$ µg m$^{-3}$) (Donahue et al., 2012). ELVOCs in the atmosphere are formed from multiple sources and chemical processes (Ehn et al., 2014; Jokinen et al., 2015). Organic mass fraction remaining (OA MFR) at 180°C measured in the roadside trailer (~10-20%) using the TD/ACSM likely provide an approximate estimate of ELVOCs (shown with green circles in Figure 2a, b). Similar values were measured during ambient TD measurements in urban background and rural sites in southeastern US (Saha et al., 2017b). Recent laboratory-derived POA volatility distributions suggest that the presence of ELVOCs in fresh

traffic-emitted POA may not be significant (May et al., 2013a, 2013b). Therefore, as a first order approximation, ELVOCs measured in the near-highway environment are likely dominated by regional background aerosol, and thus a gradient downwind of the roadway is not expected. On the other hand, traffic emissions are a major contributor of BC in near-highway environments (Baldauf et al., 2008; Bond et al., 2013; DeWitt et al., 2015) and a rapid downwind decay of BC concentrations was observed in our site (Saha et al. 2017a), consistent with past studies (Karner et al., 2010).

The downwind gradients of VFR of PM$_{0.4}$ at 180°C correlate well with that of BC (Figure 2 a, b). A less sharp decay of BC during winter was also consistent with the gradient of VFR of PM$_{0.4}$ at 180°C in winter. Figure 2c shows scatter plot of the BC fraction in PM versus VFR of PM$_{0.4}$ (at 180 °C) after subtracting OA MFR (at 180°C) measured at 10 m for the winter data set; a similar plot for the summer dataset is shown in Figure S6. These correlation analyses suggest that the observed




downwind trend of VFR of $PM_{0.4}$ at 180 °C is likely dictated by BC. BC fraction in this analysis was estimated as the ratio of measured BC concentrations from PAX to PM mass concentration from integrated volume-weighted SMPS size distribution with an estimated effective density of 1.5 g $cm^{-3}$. See Figure S7 for details on the estimation of effective density and comparison of submicron mass concentrations measured by SMPS and ACSM+PAX.

The diurnal profile of SP2-measured BC size distribution, shown in Figure S8, indicates that BC is strongly correlated with the diurnal profile of traffic volume, indicating vehicles are the major source of BC at this near-highway site. Figure 2d explores the contribution of BC to the thermodenuded SMPS size distribution at 180°C. In Figure 2d, to directly compare with the volume-weighted SMPS distribution, the mass-weighted BC size distribution was converted to a volume-weighted

distribution by assuming a BC density of 1.8 g $cm^{-3}$. The BC distribution accounts for approximately 35% of the area under the thermodenuded particle size distribution at 180 °C (Figure 2d). The remaining approximately 65% of material should consist of different low-volatility species (e.g., ELVOCs and others). This is broadly consistent with the measured OA MFR at 180°C at the roadside trailer, which explained ~50% of measured VFR at 180°C at that location (Figure 2 a, b).

### 3.2 Mixing state of near-highway particles

Figure 3 examines the mixing state of near-highway particles using V-TDMA data. The evolution of the size distribution of a monodisperse particle upon heating at different TD temperatures is referred to as a volatility spectra. Figure 3 shows the measured average volatility spectra of 25, 50, and 100 nm particles collected 10 m from the highway during summer; similar winter observations are shown in Figure S9. Heated monodisperse particles yield a bimodal size distribution; one mode (less volatile; LV mode) shows almost no change from its original diameter with heating, and the other mode (more volatile; MV

mode) shrinks substantially with heating. Similar bimodal distributions have been observed in previous near-road studies (Biswas et al., 2007; Kuhn et al., 2005; Tiitta et al., 2010). The general trend was found to be consistent across seasons.

A large fraction of LV mode particles is expected to be fresh soot from traffic emissions. Figure 3 suggests that LV mode particles are externally mixed (e.g., soot and OA exist in different particles), because if they were internally mixed with semi-volatile organics or others compounds (i.e., were coated), the coating material would evaporate with heating and a substantial

diameter reduction be observed. Several studies have shown that these LV mode particles are less hygroscopic using a V-TDMA coupled with an H-TDMA system (Kuwata et al., 2007; Tiitta et al., 2010). The presence of externally mixed LV particles was observed for all sizes studied (25, 50, 100, and 250 nm). However, the LV mode was relatively less pronounced for smaller sizes (e.g., 25 nm) compared to larger particles (e.g., 100 nm). The SP2-measured BC number size distribution peaked around 100-130 nm (see Figure 2d and Figure S8c), consistent with this observation. Kuhn et al. (2005) and Biswas et

al. (2007) also reported that the less/non-volatile fraction of near-road aerosols increased with size within the size range studied (20-120 nm) in near-road V-TDMA measurements in California.



Figure 4 explores the mixing state of near-highway BC particles using the SP2 lag-time approach (Moteki and Kondo, 2007; Schwarz et al., 2006; Shiraiwa et al., 2007; Subramanian et al., 2010). The delay time between the occurrence of scattering and incandescence peaks observed in the SP2 can be used as an indicator of relative coating thickness ($\Delta\tau = \tau$ incandescence - $\tau$ scattering = time to 'boil off' coating) (McMeeking et al., 2011a; Moteki and Kondo, 2007). Figure 4 shows the

frequency distributions (histograms) of delay time ($\Delta\tau$). Following McMeeking et al. (2011a), the entire ensemble of refractory BC-containing particles with scattering responses within detection range was considered in this analysis. Measurements are stratified by wind direction to separate those measured during wind events coming off of the highway (southwesterly; 225±45°) to the monitoring site and the opposite wind direction (northeasterly; 45±45°). Two distinct peaks near $\Delta\tau$ ~0.5 μs and ~3.5 μs appear in the $\Delta\tau$ frequency distribution. We use this to classify BC particles into two types using a threshold $\Delta\tau$ of 2 μs:

thinly coated BC ($\Delta\tau < 2\mu s$) and thickly coated BC ($\Delta\tau > 2\mu s$). The threshold criterion is based on the observed minimum in the bimodal frequency distribution of $\Delta\tau$ (McMeeking et al., 2011a; Moteki and Kondo, 2007).

Figure 4 shows that a large fraction (up to 80%) of refractory BC (rBC) containing particles at this near-highway site are thinly coated (externally mixed), and are likely fresh soot particles from traffic emissions. The observed relative proportion of thinly coated (fresh) particles increases when the wind comes off of the highway to the monitoring station (southwesterly

wind) (see Figure 4), suggesting that the local source (I-40 traffic) was the main contributor to this fraction. Using the data collected with the wind coming off of the highway, Figure S10 shows the linkage among the diurnal variation of $\Delta\tau$ frequency distributions, BC size distributions, and thermodenuded SMPS size distribution at 180°C. The thinly-coated fraction was found to be slightly higher in the midday and morning compared to the evening. This trend correlates with the diurnal variation of heavy-duty vehicle (HDV) fraction (indicated in inset of Figure S10), suggesting that HDV are the dominant contributor to the

observed fresh (thinly-coated) BC fraction. The thickly coated fraction is likely contributed by regional background aged BC particles. However, approximately 10% of fresh BC from vehicular emissions could be thickly coated as reported by Willis et al. (2016). With an opposite wind (northeasterly), a minimum direct influence from I-40 traffic is expected at our monitoring location. The observed thickly coated fraction at that wind condition went up to 41%. This range of values are consistent with past studies, For example, a thickly coated rBC fraction of approximately 30-40% is reported in previous measurements in

diverse urban environments (McMeeking et al., 2011a; Shiraiwa et al., 2007; Subramanian et al., 2010).

The substantial presence of thinly-coated (fresh) rBC suggested by the SP2 data (Figure 4) is consistent with our independently measured V-TDMA observations of externally-mixed characteristics for LV mode particles (Figure 3, Figure S9). These observations are also in agreement with several recent studies that examined the mixing state of rBC from traffic emissions using a range of techniques (China et al., 2014; Kuwata et al., 2009; Liu et al., 2014; McMeeking et al., 2011b;

Willis et al., 2016). Willis et al. (2016) reported approximately 90% of rBC mass resides in rBC-rich particles using Soot Photometers – Aerosol Mass Spectrometer (SP-AMS) measurements of traffic emissions in an urban setting, whereas the remaining 10% were mixed with hydrocarbon-like OA (HOA). China et al. (2014) reported ~72 % of soot particles from vehicle exhaust are barely or thinly coated using microscopic imaging technique. Traffic-dominated rBC particles were



reported to be uncoated or very thinly coated by Laborde et al. (2013) and Liu et al. (2014) using SP2 measurements in urban environments.

Figure 5 compares similar V-TDMA measurements of 100 nm particles at different distances from the highway to examine the evolution of mixing state of particles with downwind transport. The overall concentration of both LV and MV mode particles rapidly decreases with distance due to dilution and mixing with cleaner background air. However, LV mode particles (e.g., BC) remain mostly externally mixed at 220 m downwind distance. This result suggests that the mixing state of traffic-emitted particles is not substantially altered within a few hundred meters of the highway. Evolution of BC mixing state is typically observed in the atmosphere with photochemical aging; externally mixed BC particles (thinly coated) become progressively internally mixed (thickly coated) via formation of condensable vapors via photochemical processes followed by condensation on BC. Timescales on the order of an hour are typically required to observe a significant change in BC coating (Adachi and Buseck, 2013; McMeeking et al., 2011a; Shiraiwa et al., 2007; Subramanian et al., 2010). Since transport times of particles at 220 m downwind from the highway are on the order of a few seconds to minutes, it is not surprising to observe no significant change in mixing state of traffic-emitted particles (e.g., BC) within this short distance.

### 3.3 Inferred volatility distributions from TD data

This section discusses TD-derived volatility distributions at different distances from the highway to provide insight into the evolution of volatility of traffic emitted particles and provide parameterizations to explain phase-partitioning of near-road particles in similar microenvironments and laboratory observations. For this, we focus on the measured evaporation of ultrafine particles (25, 50, and 100 nm) at 10 and 220 m distances. Figure 6 shows measured and modeled thermograms (plots of VFR versus TD temperature) for TD measurements of varying particle sizes collected at different distances and seasons. At a particular TD temperature, smaller particles evaporate more than larger size particles (Figure 6). Size or composition (volatility distribution) may contribute to differential evaporation observed for different size particles (Saleh et al. 2011); both factors were taken into account during evaporation kinetics modeling following the framework described in Saha et al. (2015). The evaporation kinetics model tracks changes in diameter as aerosol with prescribed properties pass through the TD at a particular operating condition (T, Rt), as described in Sec 2.4; model VFR was estimated based on predicted change in particle diameter. In our fitting, we solved for particle volatility distributions (particle-phase distribution;$\{x\}_i$) via least-squares fitting of modeled and measured VFR. Fitted distributions are listed in Table S2 and model lines in Figure 6 are shown using these best-fit distributions. Our fitting results (Figure 6) show that at a particular downwind-point, a single volatility distribution can explain the observed evaporations for different sized particles, suggesting particles within this size range have a consistent volatility distribution or chemical signature. We also report $f_i$ distributions (Table 1) after converting our TD-derived particle-phase distributions ($x_i$) to total (gas+particle) distributions ($f_i$) under a gas-particle equilibrium conditions and assuming a typical near-road OA loading of ~ 5 µg m$^{-3}$ (see SI Sec. S3 for details).



Figure 7 shows simplified representations of particle volatility distributions at roadside (10 m) and downwind (220 m) locations across summer and winter seasons. In this figure, distributions of particle-phase material are shown in two broad volatility categories: extremely-low+low volatility (ELVOC+LVOC) (C* bins ≤ 0.1 µg m$^{-3}$) and semi-volatile (SVOC) (C* bins ≥ 1 µg m$^{-3}$) (Donahue et al., 2012). A laboratory-derived POA volatility distribution of gasoline vehicle exhaust by May

et al. (2013a) (derived from chromatographic analyses of filter samples) is also shown under a typical near-road aerosol loading ($C_{OA}$ = 5 µg m$^{-3}$). The volatility distributions shown in Figure 7 and also reported in Table 1 and Table S2 are at a reference temperature of 25°C, which allows a convenient side-by-side comparison across seasons and with previous studies. The gasoline POA distribution by May et al. (2013a) places ~45% of OA in the SVOC bins under this condition. Our TD-derived results show that overall volatility of near-road particles is lower than laboratory-derived POA distribution, varies across

seasons, and decreases with distance. The extracted volatility distributions of near-road particles is a mixture of traffic-emitted POA and background particles. Therefore, it is not expected that the overall volatility of near-road particles would be the same as that of vehicle POA. In Sec. 3.4, we use our spatial measurements of near-road aerosol (traffic+background) volatility to assess how/whether this laboratory-derived POA distribution can be used to represent the overall near-road volatility under real-world conditions.

Figure 7 indicates that the overall volatility of near-road aerosols decreases with distance from the highway in both seasons. For example, the TD-derived distributions apportion approximately 20-30% and 10% of particle-phase mass as SVOC at 10 and 220 m, respectively; consistent with dilution-driven evaporation of SVOCs and/or mixing with the background particles. When a volatility comparison is made at a common temperature of 25 °C (Figure 7), the particle volatility was found to be slightly higher for the winter data set than summer, especially closer to the highway. The extent of dilution and

temperature of dilution air dictate the overall partitioning of semi-volatile emissions. Atmospheric dilution was substantially lower in winter at our site (and generally) due to more stable atmospheric conditions under colder weather (Saha et al., 2017a). Therefore, when comparing particle volatility at the same temperature, lower dilution during winter likely explains the observed higher SVOCs fraction.

The volatility comparison in Figure 7 is shown at a reference temperature of 25°C; actual partitioning will vary with

ambient temperature. Under a particular $C_{OA}$ loading (atmospheric dilution), it is expected that a higher fraction of semi-volatile material partitions into the particle-phase at colder temperatures. Figure S11, an alternate version of Figure 7, shows the volatility comparison at campaign-average ambient temperatures of ~5 and ~30° C in winter and summer, respectively. After accounting for seasonal temperature difference, Figure S11 indicates that the particle-phase SVOCs fraction are approximately 2.5 times higher in winter (45%) than summer (18%) at the roadside location (10 m). During winter, a higher

fraction of semi-volatile particles may form via homogeneous nucleation during a rapid cooling of vehicle exhaust under a lower ambient temperature (Du and Yu, 2006; Kittelson et al., 2006; Kittelson and Kraft, 2014). This fact was supported by our observed three-fold increase in ultrafine particle and also HOA emission factors during winter in compared to summer,

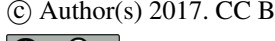



discussed in Saha et al. (2017a). This result implies that human exposure to semi-volatile particles at a near-road location could vary substantially across seasons and would be more extreme in colder weather.

### 3.4 Evaluation of laboratory-derived POA distributions to explain real-world partitioning

Particle mass concentrations measured next to a highway are a mixture of a traffic contribution and background particles. It can be reasonably hypothesized that the volatility distribution of near-road particles at a particular near-road location is a superposition of that from background particles and traffic-contributed particles at that location. One can test this hypothesis if the volatility distributions of different populations of particles (background, traffic, near-road downwind) are known. Vehicle-emitted POA volatility distributions have been derived in laboratory studies (May et al., 2013a, 2013b) of a relatively small number of vehicles in controlled laboratory tests; they have also recently been measured in a traffic tunnel study (Li et al., 2016). Here, we used our spatial measurements of particle volatility distributions along with a laboratory-derived POA distribution from May et al. (2013a) to assess our ability to represent the volatility of POA from the overall traffic fleet in a complex near-road microenvironment.

Although we cannot directly measure the traffic-contributed particles at a particular near-road location, it can be inferred from measurements, as shown in Figure 8a. Figure 8a shows an example of the measured upwind (background) and downwind organic aerosol (OA) mass concentrations as a function of distances from the highway. This example is shown from a morning transect on June 12, 2015 (summer), with the wind consistently coming off of the highway. Approximate estimates of OA mass concentrations are calculated from integrated volume of SMPS measurements and an estimated effective density of 1.5 g cm$^{-3}$ (Figure S7) and subtracting the contribution of BC (as a function of distance), nitrate and sulfate aerosols (measured by an ACSM at the near-road fixed site trailer). Finally, traffic contribution at the roadside location was approximated as the difference between concentrations measured at roadside (downwind) and upwind background location, as shown in Figure 8a.

Figure 8b shows distributions of different OA mass concentrations (near-road at 10 m, traffic contribution at 10 m, and background; as identified in Figure 8a) in volatility space. Distributions of particle-phase-only OA concentrations are shown. In this analysis, the volatility distribution of POA emissions from gasoline vehicle exhaust from May et al. (2013a) (Table 1) is assumed as representative for overall traffic-emitted OA. Our TD-derived volatility distribution at 10 m was used to distribute total OA concentrations at 10 m. Since we did not measure particle volatility at our upwind background site, we assume that the volatility distribution derived at 220 m downwind is a representative distribution for background particles. This is a reasonable approximation as particle concentrations approach background levels within 200-300 m from the highway (Saha et al., 2017a). The TD-derived distributions from the summer campaign (Table 1) were used to be consistent with the OA concentration measurements in this particular example. Overall, a good agreement was found between the measured distribution at 10 m downwind and superimposed distribution of (background+ traffic POA) (Figure 8b). In particular, the contribution from more volatile materials (C* = 1 and 10 µg m$^{-3}$) from traffic POA is required to explain the greater contribution from these more volatile species at 10 m vs. 220 m from the roadway. Therefore, this analysis indicates that





laboratory-derived volatility distribution from May et al. (2013a) can do a reasonably good job in explaining the observed partitioning of vehicle emissions in a complex near-road environment.

In addition to the analysis in Fig.8b that approximated the traffic-OA from background-subtracted roadside concentrations, the Hydrocarbon-like OA (HOA) factor derived from ACSM-measured mass spectra at roadside location can also provide an
estimate of the contribution of traffic OA (Canagaratna et al., 2010; Ng et al., 2011). A similar analysis is shown in Figure S12 using the 'coarse' tracer m/z based factor analysis approach to decompose OA mass spectra (Ng et al., 2011), where HOA factor is assumed to represent traffic-sourced OA and oxygenated-OA (OOA) background-OA. The estimated traffic-OA (HOA factor) contribution was found to be substantially lower (~5-10x) than that derived in Figure 8a based on background-subtracted roadside concentrations measured by SMPS in combination with other data. Therefore, the volatility of near-road
particles is dominated by contribution from background particles and the combined distribution does a poor job of recreating the observed near-road volatility distribution. Several factors likely contribute to this discrepancy. For one, since the correlation equations for tracer m/z based factor analysis are empirical (Ng et al., 2011), this method's accuracy and representativeness may be limited in many environments. Further, a substantial fraction of traffic-emitted smaller particles may fall outside of transmission window of ACSM (< 70 nm), but these small particles likely do not have significant effects to overall mass
contribution. On the other hand, measurements of fresh vehicle-emitted particles with an SMPS may be biased high to some extent due to the non-spherical morphology of fresh soot particles (DeCarlo et al., 2004; Maricq and Xu, 2004; Park et al., 2003). For example, effective densities of vehicle-emitted particles with mobility diameters of 200 nm may be <0.3 g cm$^{-3}$, a factor of 5 lower than that value assumed here (Maricq and Xu, 2004). The true estimates of traffic-OA likely fall between the ACSM-HOA and SMPS-based estimation. However, these two estimates may be considered as bounding cases. Further efforts
should be made to investigate the closure of estimates from different instruments and approaches.

## 4. Conclusions and Implications

Field experiments were conducted across two seasons in an effort to explore the evolution of volatility and mixing state of near-road particles within a few hundred meters downwind of a highway. The spatial distributions of the volatility of near-road aerosols varied with distance from the highway and season. The overall volatility of near-road particles decreases with
distance from the roadway. For example, at a reference temperature of 25 °C, while approximately 20-30% of particle mass was classified as semi-volatile (SVOCs; $C^* \geq 1 \mu g\ m^{-3}$) 10 m from the roadway, only ~10% of particle mass was attributed to semi-volatiles at 220 m. The decrease of the semi-volatile fraction in particle-phase with downwind distance is likely due to dilution-driven evaporation of SVOCs as fresh vehicle-emitted particles are transported downwind and/or mix with background particles. The relative abundance of semi-volatile material in the particle-phase increased during winter, especially
closer to the highway, reflecting the effect of temperature on semi-volatile partitioning. The non-volatile fraction in roadside aerosols appeared to be mostly externally mixed, and their mixing state showed minimal change within a few hundred meters from the highway.





This research has several important implications for measurement and modeling of emissions and exposure to ultrafine particles (UFPs) in a near-road microenvironment and its regulation. First, the measured particle number (PN) concentrations in near-road settings are dominated by UFPs. In a companion paper (Saha et al. 2017a), we showed that UFP number emission factors are substantially higher, and dispersion slower, during winter, indicating human exposure to UFPs would be significantly higher in colder conditions. This paper shows that a significant fraction of UFPs are semi-volatile in nature, and hence a larger portion of semi-volatile materials likely exist in particle-phase in colder conditions. Current European vehicle particle number emission standards use measurements of thermally treated exhaust that strips the semi-volatile particle components to constrain variability between measurement approaches (Wang et al., 2017). As a result, this regulatory measurement likely does not address the seasonally- and spatially-varying real-world particle number concentrations and compositions to which people are exposed. Several recent toxicology studies (Biswas et al., 2009; Keebaugh et al., 2015) reported that the semi-volatile species in traffic-sourced particles could be more toxic than less volatile components. Therefore, our observed seasonal variation in UFPs emission factors and semi-volatile components in the particle-phase suggest that human exposure to UFPs and its toxicity in a near-road microenvironment could vary with seasons and environmental conditions and would be more extreme in colder weather. The elevated fraction of semi-volatile materials in roadside particles and their potential higher toxicity suggest that an equivalent amount of exposure (concentration × duration) to roadside versus background particles could have significantly different health impacts. However, the toxicity of different volatility and size classes of PM is not well established in the current literature. For example, Cho et al. (2009) reported no significant difference in the overall toxicity end points for PM samples collected at 20 m and 275 m from an interstate highway. Further research is needed to better understand toxicity and health impacts of different volatility and size classes of PM from different sources and environmental conditions. Second, our finding of externally mixed near-road particles suggests that exposure to BC and OA containing particles could be different across seasons. For example, OA containing particles will be more dynamic under changing ambient conditions. Environmental conditions (temperature, atmospheric dilution) will influence the gas-particle partitioning of SVOCs, thus exposure to condensed- versus vapor-phase SVOC under changing ambient conditions. On the other hand, exposure to BC would be less influenced by changing ambient conditions. Finally, the volatility distributions and mixing state characteristics of near-road particles derived here can be used to examine the representativeness of laboratory derived results in complex real-world scenario (as shown via an example in this paper) and to improve the representation of traffic-sourced aerosols in air quality, exposure assessment and chemical transport models.

## Acknowledgements

We thank Sue Kimbrough, Rich Baldauf and Richard Snow of the US EPA Office of Research and Development, Research Triangle Park (RTP), NC for help and support with establishing the roadside trailer facility for this study. We thank Steve Weiandt (Atlantic Investment Management) for providing access and support with establishing the background site for this study. We thank Nagui Rouphail and his research group (NCSU) and NC-DOT for the traffic data used in this paper.



Funding was provided by the Health Effects Institute (HEI) under RFA 13-1. Contents of this paper are solely the responsibility of the authors and do not necessarily represent the official views of HEI, and no official endorsement should be inferred.

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



**Table 1:** TD-derived particle volatility distributions (at 298 K) at 10 m and 220 m downwind distance from the highway I-40 during summer and winter. Laboratory-derived gasoline POA distribution by May et al. (2013a) is also listed.

| $\log C^*$ at 298K | [a]TD-derived $f_i$ distribution | | | | [b]Gasoline POA (May et al. 2013) |
|---|---|---|---|---|---|
| | 10 m (Summer) | 220 m (Summer) | 10 m (Winter) | 220 m (Winter) | |
| -4 | 0.07 | 0.10 | 0.18 | 0.28 | |
| -3 | 0.13 | 0.21 | 0.07 | 0.08 | |
| -2 | 0.16 | 0.20 | 0.14 | 0.20 | 0.14 |
| -1 | 0.27 | 0.37 | 0.15 | 0.30 | 0.13 |
| 0 | 0.12 | 0.06 | 0.27 | 0.09 | 0.15 |
| 1 | 0.25 | 0.06 | 0.20 | 0.06 | 0.26 |
| 2 | | | | | 0.15 |
| 3 | | | | | 0.03 |
| 4 | | | | | 0.03 |
| 5 | | | | | 0.01 |
| 6 | | | | | 0.11 |

[a]TD-fitted particle-phase distributions ($x_i$) with $\gamma_e = 0.25$ and $\Delta H_{vap} = 100$ KJ mol$^{-1}$ (reported in Table S2) are converted to total (gas+particle) distribution ($f_i$) under gas-particle equilibrium condition and assuming a total aerosol loading of ~ 5 µg m$^{-3}$. (Conversion equations are given in S1, Sec. S3).

[b]Chromatographic analysis



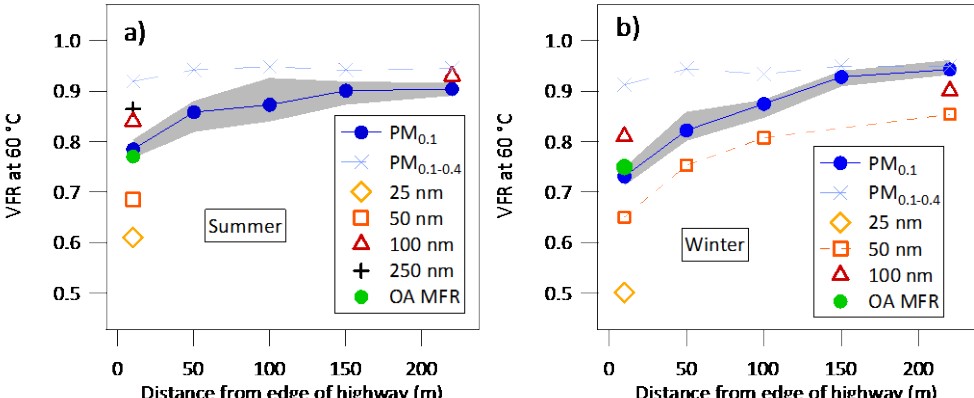

**Figure 1:** Campaign-average downwind evolution of volume fraction remaining (VFR) of near-road particles after heating at 60 °C in a TD (Rt = 30 s) during (a) summer, and (b) winter. VFR of size-selected particles (e.g., 25, 50, 100, 250 nm) obtained from V-TDMA measurements. VFR of $PM_{0.1}$ and $PM_{0.1-0.4}$ estimated from integrated SMPS volume between 10-100 nm and 100-400 nm, respectively. OA MFR was measured using a TD/ACSM system (Rt = 30 s) in the near-road trailer. The shaded area represents interquartile range of all measurements for each season.





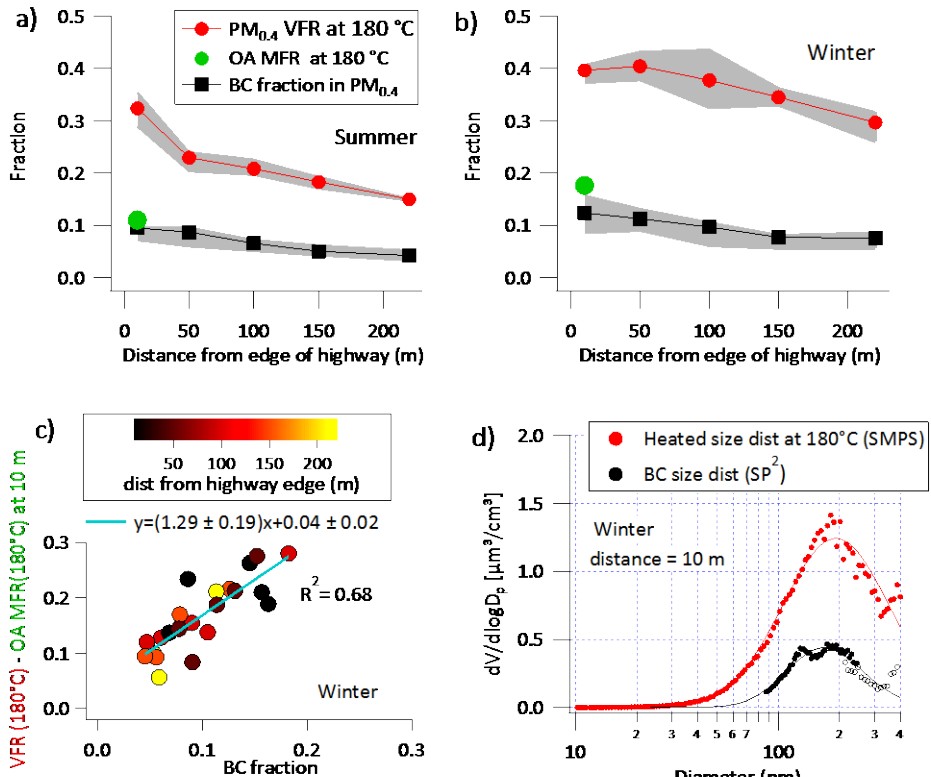

**Figure 2:** Campaign-average downwind evolution of volume fraction remaining (VFR) of $PM_{0.4}$ (integrated SMPS volume over 10-400 nm) at 180 °C in a TD (Rt = 30 s) and black carbon (BC) fraction in $PM_{0.4}$ during (a) summer, and (b) winter. Points are mean and the shaded area represents interquartile range. OA MFR was measured using a TD/ACSM system (Rt = 30 s) only in the near-road trailer. (c) Correlation between the BC fraction and VFR of $PM_{0.4}$ (at 180 °C) at various downwind distances after subtracting OA MFR (at 180°C) measured at 10 m. (d) Comparison of SMPS-measured thermodenuded size distribution at 180 °C and SP2-measured BC size distribution.





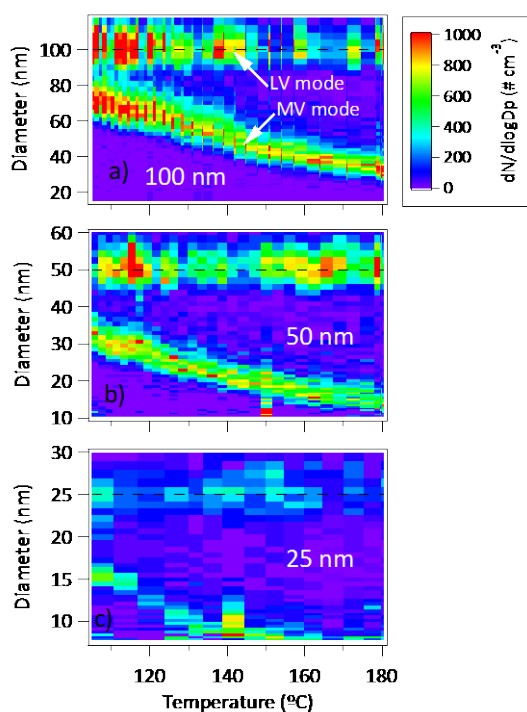

**Figure 3:** Average V-TDMA volatility spectra of 25, 50, and 100 nm particles collected 10 m from I-40 during summer. Figure S9 shows similar plots for the winter data set.



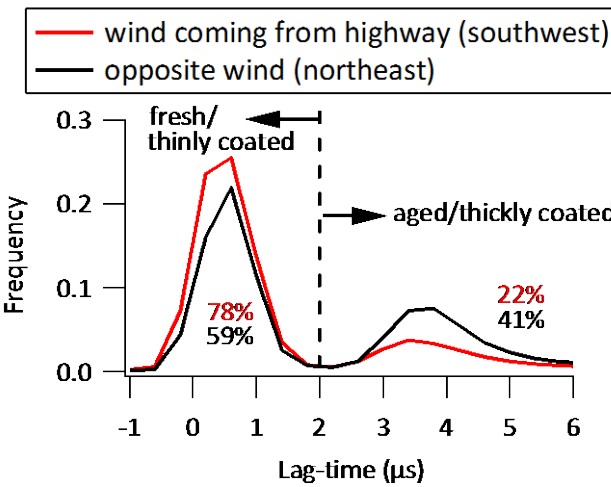

**Figure 4:** Campaign-average frequency distributions (histograms) of SP2 lag-time ($\Delta\tau$) for refractory BC-containing particles measured during periods with winds from the highway (red) or from the opposite direction (black). Measurements were collected at 10 m distance from the highway in winter.





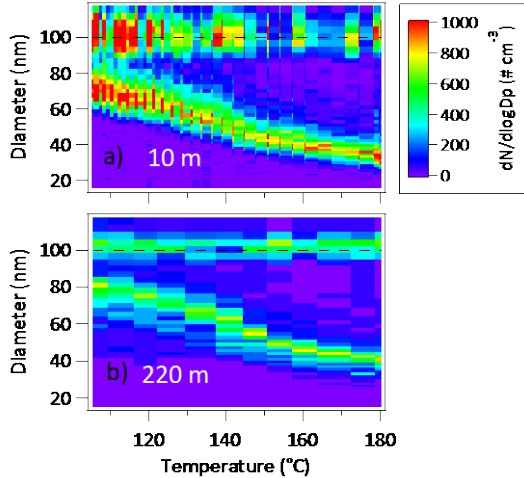

**Figure 5:** Similar to Figure 3 showing average volatility spectra of 100 nm particles at 10 m and 220 m downwind of the highway. Measurements were collected during transect runs in summer.





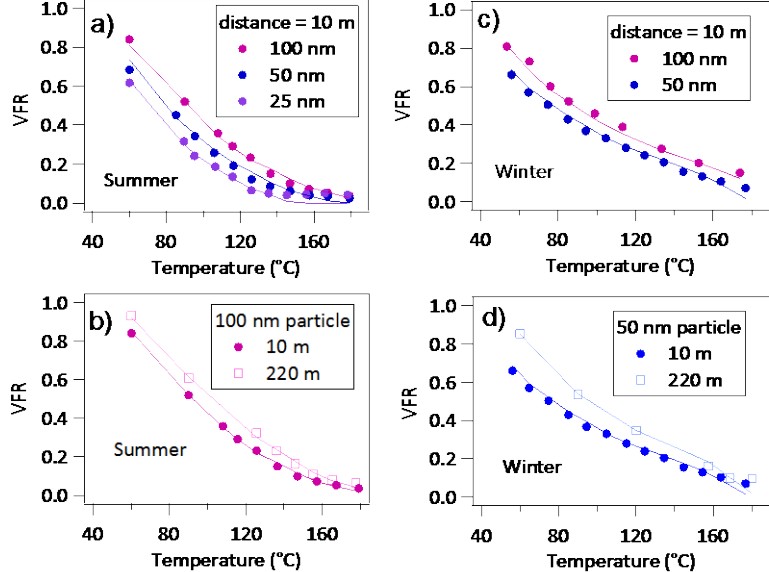

**Figure 6:** Campaign-average measured (points) and modeled (line) thermograms for different sized particles measured at 10 m and 220 m downwind during summer (a-b) and winter (c-d). Model lines are shown using the best fitted volatility distributions listed in Tables 1 and S2, and $\Delta H_{vap}$ = 100 KJ mol$^{-1}$ and $\gamma_{e}$ = 0.25.





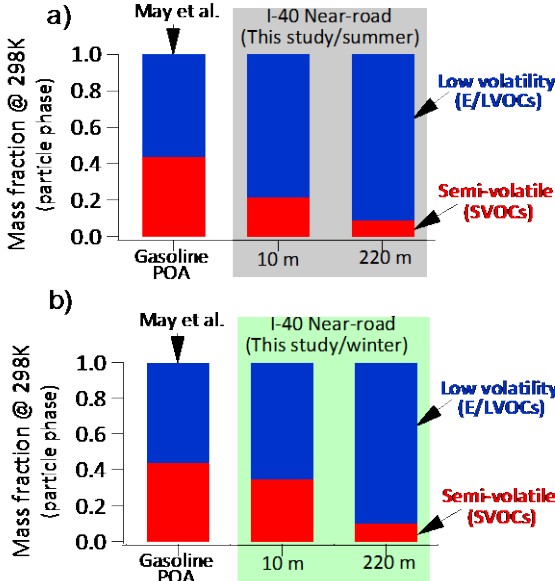

**Figure 7:** Comparison of volatility classification of near-road particles measured at 10 m and 220 m (this study) for a) summer, and b) winter at a reference temperature of 25ºC. Distributions of particle-phase material are shown using two broad volatility categories. Also shown in both panels is the POA distribution from gasoline vehicle exhaust by May et al. (2013a) under typical near-road aerosol loading ($C_{OA}$~5 µg m⁻³).




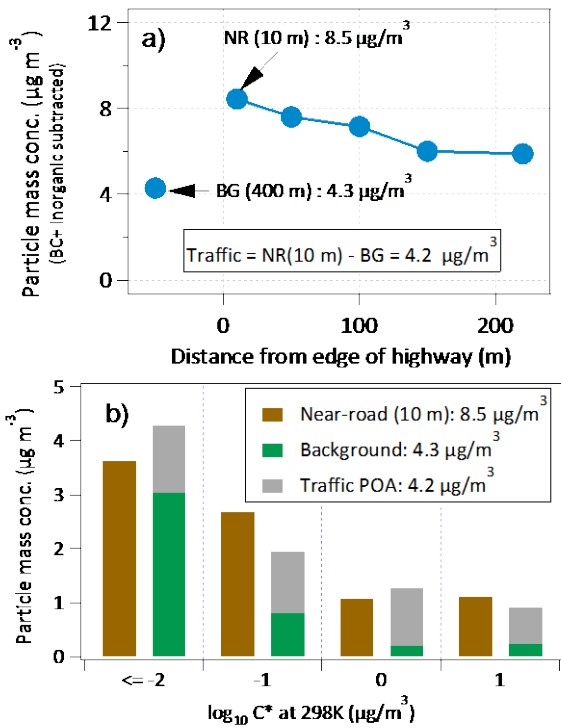

**Figure 8:** (a) An example of measured upwind (background) and downwind concentrations of OA mass loading as a function of distances from the highway. (b) Distributions of OA mass loading (particle-phase only) as measured at 10 m downwind (brown bar) and apportioned to vehicle emissions (grey) and background (green) volatility distributions using distributions from May et al. (2013a) and fits to downwind (220 m) TD data, respectively.