# Peer review of "Downwind evolution of the volatility and mixing state of near-road aerosols near a US interstate highway"

_Atmospheric Chemistry and Physics, 2017_

## Referee Comment (RC1) · Anonymous Referee #1 · 4 Oct 2017

In this study, the authors report composition, volatility and mixing state of traffic-related aerosols measured near a highway, 10 m to 220 m from the road edge. Measurements were conducted under varied environmental conditions including winter and summer campaigns. The investigation is important because vehicle emissions undergo a rapid transformation in ambient air which influences the key properties of combustion emissions with respect to their health and environmental effects. The data provided by this work is valuable and data-analysis is ambitious. Especially, this manuscript offers a high-quality data about the volatility distribution of traffic emissions. The topic is fitting well in the scope of ACP. Revised manuscript is suitable for publication in ACP after following comments are addressed:

[Figure]

- The last paragraph of chapter 3.4 (tracer m/z based factor analysis) should be shortened. Perhaps, this paragraph could be moved to the supplement. Did you use estimation HOA = 13.4 × (C57 - 0.1 × C44) (Ng et al., 2011) ? I would recommend using PMF (Ulbrich et al., 2009) instead of tracer-based factor analysis, if possible.

- Figure 8: Figure 8 is quite hard to read and should be improved e.g. by adding summer and winter volatility distributions (Table 1) to Fig. 8b for comparison and clarify figure caption. If bins 2-6 (Table 1) are not applied in this work, please, consider removing these bins from Table 1 (or add marking for these).

- Fig S3: Size distributions of nucleation mode particles look odd. Please, check SMPS data carefully, especially particles smaller than 10 nm.

- Fig S7: AN and AS mass concentrations (calculation) based on a statement that aerosols are neutral. Please, add acidity plot (NH4 measured vs. NH4 calculated) to prove that assumption (Supplement) or use model such as Aerosol Inorganic Model II (AIM-II) (Clegg et al., 1998) for acidity calculation. An effective density calculation (Kuwata et al., 2011) of OA is limited to particle components having negligible quantities of additional elements. How traffic emissions components such as black carbon and NO3 effect on density calculations?

References

Clegg, S. L., Brimblecombe, P., and Wexler, A. S.: A thermodynamical model of the system H-NH4+ -SO42–NO3− -H2O at tropospheric temperatures, J. Phys. Chem. A, 102, 2137–2154, 1998.

Ulbrich, I. M., Canagaratna, M. R., Zhang, Q., Worsnop, D. R., and Jimenez, J. L.: Interpretation of organic components from Positive Matrix Factorization of aerosol mass spectrometric data, Atmos. Chem. Phys., 9, 2891–2918, doi:10.5194/acp-9-2891-2009, 2009.

[Figure]

2017.

---

## Referee Comment (RC2) · Anonymous Referee #2 · 19 Oct 2017

The authors present a dataset of aerosol evaporation in thermodenuders and ACSM and black carbon measurements measured at several distances from a major highway. The dataset and the associated analysis of the volatility and the mixing state of the aerosol at different distances downwind of the road is very valuable to atmospheric aerosol and traffic emission researchers. The measurements are to my knowledge the first time the volatility and mixing state have been measured at several points along the roadside, and as such the dataset is novel and interesting, and the subject area is clearly in line with ACP. The manuscript is well written and the data analysis is comprehensive; there are some issues that I suggest are addressed in a revision, after which I recommend publication.

1. As the traffic-originated aerosol is transported away from the source, it dilutes with background aerosol with a rate that depends on the wind velocity and the atmospheric stability. Therefore, the same measurement point (measured by distance) can see aerosol with different age since emission, and different dilution factors. This can be be overcome by measurements of some relatively inert gas at the roadside and downwind, or more crudely just estimating the aerosol 'age' from the wind velocity, distance (and direction).

Neither of these have been performed in the manuscript, and therefore the measurement at each distance may include differently aged particles, with a varying fraction of background aerosol mixed in the traffic-originated aerosol. This impacts the generalization of the results (which may easily happen, given the scarcity of this type of data in the literature), and I think that this should be made clear to readers. This could be done e.g. by providing estimates, and ranges of variation, of the age of particles as an additional variable. Also, this would give more confidence in whether the observed seasonal differences are due to actual differences in volatility or maybe just different mixing situations.

2. Comparing the distributions in Fig S3, it seems that an important factor in the change of volatility is the disappearance of a large fraction of particles smaller than ca 40-50 nm between the measurement points at 10 m and 150 m, and this seems to contibute to the large change in volatility of <100 nm particles (the change seems much less for the 100-400 nm particles). From this one could assume that these small particles are mostly consisting of SV particles. On the other hand, these smallest particles are most affected by e.g. coagulation, and their mass could thus be transferred to the larger size range during transport. I think that the large change with transport in the <50 nm particle size range should be investigated, and maybe some overview of the literature on whether this is a typical occurrence could be added to the paper.

Some other minor comments

page 8, line 15: 'Other effects (if any)...' This sentence is mostly confusing; either the other effects are clearly mentioned here, or this sentence could be removed.

page 9, line 1: I'm not following what is meant by 'downwind trend is dictated by BC'. What is the BC-related process that determines the VFR? Also, it is not clear what the quantity VFR-OA is representing here. How is this related to the difference in BC and denuded SMPS volume in Fig. 2d? Also, is fig 2d an example or representative of a longer period? Please clarify.

Page 9, line 15, and Fig 3 and 5. The sentence 'The evolution of the size distribution of a monodisperse particle upon heating' is confusing. Also, how is the color scale chosen in Figs. 3 and 5? As the main information of the figure is the relative contributions of the different modes, could it not be useful to normalize the volatility spectra? Now the 25 nm spectra are very vaguely readable.

page 11, line 6: '...the mixing state of traffic-emitted particles is not substantially altered within a few hundred meters'. This is basically correct in the context that particles that are externally mixed at the start stay so; however, in a hypothetical case that internally mixed particles dominate the emission at the start, the mixing with background air would soon cause an external mixture. The sentence should be made less general, e.g. '...in this case, the mixing state...'

page 13, line 26, and Fig. 8. I have some difficulties understanding how Fig. 8b was arrived at, and what should be compared in the figure. Which bars at at 10m and which ones at 220? The statement 'we assume that the volatility distribution (...) at 220 is a representative (...) for background particles' seems contradicting to fig 8a, where ca 33% of the total aerosol seems to still be traffic-originated. I think that it would be helpful to understanding if a more detailed (step-by-step) explanation for how each of the three mass concentrations were obtained could be given, maybe even in equation form in the supplementary. For example:

* M(near-road)$_i$ = (SMPS total mass - BC - ACSM(inorganics) ) x (May et al)$_i$

* M(background)$_i$ = (bg OA mass ) x (TD vol. distribution at 220m)$_i$

* M(traffic)$_i$ = (NR - BG oa mass) x (TD vol. distribution at 10 m)$_i$

Several assumed densities could be found in the article for different data analysis approaches (at least 1.1, 1.5 and 1.8 g/cm$^3$ are described). Could this be made more consistent, or are there specific reasons for using these values that could be stated?

---

## Author Response (AR1)

Response to reviewers comments for the paper "**Downwind evolution of the volatility and mixing state of near-road aerosols near a US interstate highway**" by Provat K. Saha et al.

To Whom It May Concern:

We would like to thank the two reviewers for their thoughtful and helpful comments. In addressing these comments, we feel we have substantially improved the manuscript.

Please find below detailed responses shown in **blue text** (with direct quotes from the revised manuscript shown in *italics*) to the comments and suggestions offered by the two reviewers. The changes made in the revised manuscript and SI are marked-up with blue text as well.

Best regards,

The Authors

Anonymous Referee #1

In this study, the authors report composition, volatility and mixing state of traffic-related aerosols measured near a highway, 10 m to 220 m from the road edge. Measurements were conducted under varied environmental conditions including winter and summer campaigns. The investigation is important because vehicle emissions undergo a rapid transformation in ambient air which influences the key properties of combustion emissions with respect to their health and environmental effects. The data provided by this work is valuable and data-analysis is ambitious. Especially, this manuscript offers a high-quality data about the volatility distribution of traffic emissions. The topic is fitting well in the scope of ACP. Revised manuscript is suitable for publication in ACP after following comments are addressed:

We thank the reviewer for the encouraging comments.

The last paragraph of chapter 3.4 (tracer m/z based factor analysis) should be shortened. Perhaps, this paragraph could be moved to the supplement. Did you use estimation HOA = 13.4 × (C57 - 0.1 × C44) (Ng et al., 2011)? I would recommend using PMF (Ulbrich et al., 2009) instead of tracer-based factor analysis, if possible.

Following the reviewer's suggestion, we revisited the last paragraph of section 3.4. The details on tracer m/z based analysis and the corresponding discussion are moved to the supplement (see Sec. S5: Tracer m/z based factor analysis of ACSM dataset) and this analysis is now only briefly mentioned in the main text. As the reviewer inferred, tracer m/z based OA components are estimated following Ng et al. (2011) as: hydrocarbon-like OA (HOA ~ $13.4 \times (C_{57} - 0.1 \times C_{44})$) and oxygenated OA (OOA ~ $6.6 \times C_{44}$), where $C_{57}$ and $C_{44}$ are the equivalent mass

concentration of tracer ion m/z 57 and 44, respectively. Previous evaluation of this method has shown that it can reproduce the HOA and OOA concentrations to within ~30% of the results from detailed PMF analysis (Ng. et al. 2011). A detailed PMF analysis of this data set would require an extensive new analysis and thus we considering it beyond the scope of this manuscript. Further, since the estimated traffic-OA (HOA factor) contribution was found to be ~5-10x lower than that derived based on background-subtracted roadside concentrations measured by SMPS, we do not expect that a detailed PMF analysis will close this gap. We have discussed several factors that likely contribute to this discrepancy.

Revision:  Please see revised SI, Sec. *S5: Tracer m/z based factor analysis of ACSM data set.*

Figure 8: Figure 8 is quite hard to read and should be improved e.g. by adding summer and winter volatility distributions (Table 1) to Fig. 8b for comparison and clarify figure caption. If bins 2-6 (Table 1) are not applied in this work, please, consider removing these bins from Table 1 (or add marking for these).

We thank the reviewer for paying special attention to Fig. 8, and we agree that some clarification is needed. One thing requiring clarification is that the purpose of the analysis in Fig. 8 is *not* to show the volatility distributions from summer and winter. We show simplified versions of the volatility distributions from summer and winter at different distances from the highway in Fig. 7 and the full distributions are listed in table 1. The purpose of the analysis resulting in Fig. 8 was to examine how well a laboratory-measured volatility distribution of traffic POA (May et al., 2013) can explain the observed partitioning of vehicle emissions in a complex near-road environment. In this analysis, we hypothesized that the volatility distribution of road-side OA results from a superposition/weighted average of the volatility distributions of traffic-contributed OA and background OA.  We compare our measured road-side volatility distribution (at 10 m) with a reconstructed distribution using the volatility distribution from laboratory vehicle measurements described in May et al. (2013) and the 'background' distribution measured at our furthest-from-road location (220 m). In the revised manuscript, we tried to clarify the discussion of Fig. 8 by changing the figure caption and legend, adding markers to the figure, improving the text description and adding a new supporting section in SI (sec. S4) that describes the analysis approach step-by step.

Revision: Please see (i) revised Fig. 8, (ii) revised text in Sec. 3.4 and (iii) new addition of Sec. S4 in SI

Fig S3: Size distributions of nucleation mode particles look odd. Please, check SMPS data carefully, especially particles smaller than 10 nm

We thank the reviewer for taking a close look at this Figure and pointing out this issue.  Fig.S3 showed example data from a typical transect measurement to demonstrate the relative change in particle number distributions at different distances from the highway. Data were collected using an SMPS system with a TSI 3010 CPC, which has a detection limit (lower size cut) of 10 nm. Therefore, counting for particles smaller than 10 nm was distorted and should have been excluded from the original figure. In revision, we show a similar example data set from another

transect run with data below 10 nm excluded. We mainly made this change to include an example that more clearly demonstrates, in a relative sense, the strong spatial gradient of PN size distribution above 10 nm.

Revision: Please see revised Figure S3

- Fig S7: AN and AS mass concentrations (calculation) based on a statement that aerosols are neutral. Please, add acidity plot (NH4 measured vs. NH4 calculated) to prove that assumption (Supplement) or use model such as Aerosol Inorganic Model II (AIM-II) (Clegg et al., 1998) for acidity calculation. An effective density calculation (Kuwata et al., 2012) of OA is limited to particle components having negligible quantities of additional elements. How traffic emissions components such as black carbon and NO3 effect on density calculations?

An acidity plot (of measured vs. predicted $NH_4$) has been added as Fig. S7, panel (b). The NH4 predicted is estimated as:  NH4 predicted = 2*(18/98)*SO4 + (18/63)*NO3 + (18/35)*Cl, where the fractional amounts correspond to the molecular weights of the relevant species. An ammonium balance can provide insights into the validity of assumption of neutral aerosol. Since the acidity plot (of measured vs. predicted NH4) has an average slope closer to ~1 (1.1±0.03), indicating an ammonium balance. Therefore, assumption of neutral aerosols at the measurement location was reasonable.

We agree that it is important to note the limitations associated with the application of the Kuwata et al. (2012) parameterization for OA density, which was developed based on laboratory data with negligible BC or nitrogen content. However, we are unable to comment on the extent to which this deviation will affect the density calculated in this way. We do note that Kuwata et al. applied their parameterization to data from the AMAZE campaign, which had an average OA fraction of 0.8 (versus 0.74 for our data), and found that the results agreed well with their measured density. Therefore, and given that we found the overall OA mass at this site to be dominated by relatively oxidized (OOA-like) spectra, we suspect that this parameterization does a reasonably good job describing our OA density. Therefore, we do not expect that the use of the Kuwata et al. (2012) parameterization, which assumes particle components having negligible quantities of additional elements, would introduce any dramatic bias. As noted elsewhere, we calculated the overall effective density of PM by weighting fractional contribution of major PM species (OA, BC, AN, and AS) with their respective densities; effective density of submicron PM = dens of OA $\times f_{OA}$ + dens of BC $\times f_{BC}$ + dens of AN $\times f_{AN}$ + dens of AS $\times f_{AS}$. The campaign average fractional contributions are OA ~74%, AS ~13%, AN ~7%, and BC ~6%. The density of OA is calculated using the Kuwata et al. (2012) parameterization, which depends on the molecular composition of OA (O:C, H:C).  OA was the dominant component of measured PM (~ 75%) and overall contribution of traffic emitted inorganic components (~ BC) was relatively small (BC~ 6%). Finally, since our mass comparison based on application of this density to SMPS-measured volume and ACSM+BC mass showed good agreement (Fig. S7a), this indicates our overall estimated effective density, on which OA density has the largest influence, is well constrained.

Revision: Please see revised Figure S7 and text in caption.

Anonymous Referee #2

The authors present a dataset of aerosol evaporation in thermodenuders and ACSM and black carbon measurements measured at several distances from a major highway. The dataset and the associated analysis of the volatility and the mixing state of the aerosol at different distances downwind of the road is very valuable to atmospheric aerosol and traffic emission researchers. The measurements are to my knowledge the first time the volatility and mixing state have been measured at several points along the roadside, and as such the dataset is novel and interesting, and the subject area is clearly in line with ACP. The manuscript is well written and the data analysis is comprehensive; there are some issues that I suggest are addressed in a revision, after which I recommend publication.

We thank the reviewer for their encouraging comments

1. As the traffic-originated aerosol is transported away from the source, it dilutes with background aerosol with a rate that depends on the wind velocity and the atmospheric stability. Therefore, the same measurement point (measured by distance) can see aerosol with different age since emission, and different dilution factors. This can be overcome by measurements of some relatively inert gas at the roadside and downwind, or more crudely just estimating the aerosol 'age' from the wind velocity, distance (and direction). Neither of these have been performed in the manuscript, and therefore the measurement at each distance may include differently aged particles, with a varying fraction of background aerosol mixed in the traffic-originated aerosol. This impacts the generalization of the results (which may easily happen, given the scarcity of this type of data in the literature), and I think that this should be made clear to readers. This could be done e.g. by providing estimates, and ranges of variation, of the age of particles as an additional variable. Also, this would give more confidence in whether the observed seasonal differences are due to actual differences in volatility or maybe just different mixing situations.

It is true that meteorological conditions (e.g., temperature, dilution) affect the transport and transformation of traffic plume in a near-road environment, thus can affect the volatility of transported particles. These effects were visible in our volatility measurements and extracted volatility distributions from two seasons, as discussed in Sec. 3.1 (page 8, L 5-15) and Sec. 3.3 (Page 12, L17-25). In a companion paper (Saha et al., 2017), we discussed the climatological aspects and transport of traffic plumes in detail. We observed a substantial seasonal difference in downwind decay profile of $NO_x$ and BC (traffic tracer), and so we are able to comment on the relative difference in dilution and 'age' of sampled particles. In this paper we use BC as a 'tracer' in this way in several places (e.g. Fig. 2 and related discussion). A modified version of a figure from Saha et al. (2017) with a number of 'tracers' measured near the road (NOx, and BC) is shown below for reference. *Within* a given season, we did not observe a substantial day-to-day variability in pollutant profile, possibly because we conducted our transect measurements on a few days selected based on favorable weather conditions (with the wind coming off of the roadway). Since we discussed the different climatological aspects of our site and data in our

companion paper (which will hopefully be published soon), we did not include it here. However here, and in the manuscript, we provide a basic description.

Winter measurements typically showed less rapid dilution (more stable conditions, lower wind speeds) and lower temperatures than the summer. Therefore, decay and mixing of the traffic plume was substantially slower in winter than in summer, as demonstrated by less steep downwind decay profiles of NOx and BC (traffic tracer) in winter than summer (see Fig. below). We observed the effects of these differences (temperature, dilution) in volatility distributions derived from the two seasons' data, as discussed in Sec. 3.3.

For example, we mention in Sec. 3.3 (page 12, L23-25). "*Atmospheric dilution was substantially lower in winter at our site (and generally) due to more stable atmospheric conditions under the colder weather (Saha et al., 2017a). Therefore, when comparing particle volatility at the same temperature, lower dilution during winter likely explains the observed higher SVOCs fraction.*"

In reality, the combined effects of temperature, dilution, and changes in tailpipe emission properties likely dictate the observed inter-seasonal differences in particle volatility. Although a complete decoupling each of these effects is challenging, our analysis indicates that the temperature effect is more important than others on the volatility distribution (e.g. Figs. 7, S11). The impact of dilution is less dramatic than temperature (Fig. 7 vs. Fig. S11) and a seasonal difference in tailpipe emission properties is likely also less important than the influence of ambient temperature on emission partitioning.

[Figure]

**Figure** showing **c**ampaign-average normalized downwind decay profile for a) NOx, b) BC during summer and winter.

2. Comparing the distributions in Fig S3, it seems that an important factor in the change of volatility is the disappearance of a large fraction of particles smaller than ca 40-50 nm between the measurement points at 10 m and 150 m, and this seems to contribute to the large change in volatility of <100 nm particles (the change seems much less for the 100-400 nm particles). From this one could assume that these small particles are mostly consisting of SV particles. On the other hand, these smallest particles are most affected by e.g. coagulation, and their mass could thus be transferred to the larger size range during transport. I think that the large change with transport in the <50 nm particle size range should be investigated, and maybe some overview of the literature on whether this is a typical occurrence could be added to the paper.

We agree with the reviewer that other processes, including coagulation, should be considered as possibly important influences on near-road aerosol size distribution. Zhang et al. (2004) reported that after the initial stage of dilution (tailpipe to on-road), the second phase (road-to-ambient) aerosol size distribution evolution is dominated by condensation and dilution, while coagulation and deposition play minor roles. Choi and Paulson (2016) reported that only 5-10% of particle number concentration loss could be attributed to coagulation within 200 m from the roadway. Therefore, based on the evidence from existing literature, we do not expect that coagulation is a primary process in altering the physio-chemical properties of transported traffic particles within the near-road domain (10-200 m from the highway edge) where we conducted our measurements.

Revision: We included the following discussion (Page 7 L30 -Page 8 L3):

*"It can be noted here that other processes, including coagulation, can possibly have important influences on the evolution of near-road aerosol size distribution. However, Zhang et al. (2004) reported that after the initial stage of dilution (tailpipe to on-road), the second phase (road-to-ambient) aerosol size distribution evolution is dominated by condensation and dilution, while coagulation and deposition play minor roles. Therefore, we do not expect that coagulation is a dominant process in altering the physio-chemical properties of transported traffic particles within the near-road domain (10-200 m from the highway edge) where we conducted our measurements."*

As we discuss in Sec. 3.3 and Fig. 6, our results show that a single volatility distribution can explain the observed evaporation of different size particles at a given location for the size range we investigated (25-100 nm). Therefore, our analysis suggests that particles within this size range (25-100 nm) have a consistent volatility distribution or chemical signature, although they showed a different degree of evaporation in TD. This will also be true of evaporation due to atmospheric dilution. Typically, it is expected that very small particles will evaporate more quickly due to the Kelvin effect and increased surface/volume ratio, even if they have similar volatility distribution.

page 8, line 15: 'Other effects (if any)...' This sentence is mostly confusing; either the other effects are clearly mentioned here, or this sentence could be removed.

Here, by 'other effects (if any)' we mean if there is any inter-seasonal differences in volatility. Revision: we revised this sentence as: "*Therefore, inter-seasonal differences in volatility of emissions (if any) should be reflected in the resulting fitted volatility distributions; these modeling results are discussed in Sec. 3.3.*" (Page 8, L18-19)

page 9, line 1: I'm not following what is meant by 'downwind trend is dictated by BC'. What is the BC-related process that determines the VFR? Also, it is not clear what the quantity VFR-OA is representing here. How is this related to the difference in BC and denuded SMPS volume in Fig. 2d? Also, is fig 2d an example or representative of a longer period? Please clarify.

We have made some minor edits to text in an effort to clarify this point. We'll elaborate here, but hope the modified text conveys the point more clearly. We meant here that downwind evolution of VFR at 180 C follows the similar trend as BC. One approximation we can make that VFR at 180 C ~ ELVOC (extremely low-volatility organics) + BC. Since ELVOCs (~ OA MFR at very high temperature) are likely dominated by regional background aerosol, a gradient in the contribution from these species downwind of the roadway is not expected. Therefore, we would expect that near-road spatial gradient of PM VFR at a very high temperature will be dictated by the BC component. Our analysis in Fig. 2c supports this assumption: the scatter plot of [VFR at 180 C – ELVOC] vs. BC fraction has an average slope of 1.29±0.19. An alternative way of saying this is that the difference between high-temperature-TD-processed SMPS volume and BC should give us an estimate of ELVOCs. Fig. 2d indicates that this difference is ~ 65% of denuded SMPS volume at 180 C, and thus this fraction of PM volume may be considered as ELVOCs. This is consistent with our other measurement approach shown in Fig, 2b, showing that the OA MFR at 180 C measured via the ACSM at the roadside trailer is ~ 50% of PM VFR at 180 C. Therefore, by combining measurements from different instruments and approaches, the analysis in Fig. 2 indicates that in a near-road environment, denuded particle volume at very high temperature (180 C) can be approximated as~ ELVOCs + BC, and its spatial trend is dictated by BC component (traffic tracer), not changes in ELVOC content away from the roadway. All plots in Fig.2 show campaign-average observations (as mentioned in the caption), so these trends should be fairly representative.

Revision: To summarize and clarify the analyses is Fig 2, we added the following sentences (Page 9, L16-19): "*Therefore, by combining measurements from different instruments and approaches, the analysis summarized in Fig.2 indicates that in a near-road environment, denuded particle volume at very high temperature (at 180 C) can be approximated as~ ELVOCs + BC, with a spatial trend that is dictated by BC (as a traffic tracer).*"

Page 9, line 15, and Fig 3 and 5. The sentence 'The evolution of the size distribution of a monodisperse particle upon heating' is confusing.

Revision: We revised this as: "*The heated size distributions of a size-selected monodisperse (at ambient temperature) aerosol at different TD temperature is referred to as a volatility spectra.*" (Page9, L21-22)

Also, how is the color scale chosen in Figs. 3 and 5? As the main information of the figure is the relative contributions of the different modes, could it not be useful to normalize the volatility spectra? Now the 25 nm spectra are very vaguely readable

The reviewer correctly points out that the primary information of the figure is the relative contributions of the different modes, and these data could be presented in a normalized form. However, we aimed to show the relative abundance of different size particles at a particular location, as well as how this changes with distance (especially for Fig. 5), and so choose to show absolute concentrations with the same color scale for all panels in Fig. 3 and 5.  For example, the color scale in Fig. 5 allows us to show that concentrations of 100 nm particles reduced with distance, but their mixing state remained almost unchanged. We feel this is the best overall approach for displaying these data.

page 11, line 6: '...the mixing state of traffic-emitted particles is not substantially altered within a few hundred meters'. This is basically correct in the context that particles that are externally mixed at the start stay so; however, in a hypothetical case that internally mixed particles dominate the emission at the start, the mixing with background air would soon cause an external mixture. The sentence should be made less general, e.g. '...in this case, the mixing state...'

We thank the reviewer for the suggestion. One thing to make clear here is that we state that the mixing-state of traffic particles at near-road (10 m) and far-road (200 m) remained almost same, but we didn't indicate their degree of mixing. For example, if some percentage of BC particles from traffic emissions is externally mixed at the near-road location (10 m), their mixing state did not substantially alter within a few hundred meters. If traffic particles are internally mixed at the start (e.g., BC mixed with HOA), it will remain so during mixing with background air (since that already has some mixing/coating of OA). Essentially, the fraction of internally and externally mixed particles doesn't appear to change, nor is there evidence of substantial coating on externally mixed 'non-volatile' particles.

Revision: To make this sentence more specific, we revised it as: "*This result indicates that there is minimal change in the mixing state in traffic-emitted particles between the near-road (~10 m) and far-road (~220 m) locations. Specifically, the proportion of internally versus externally mixed particles doesn't appear to change, nor is there evidence of substantial coating on externally mixed 'non-volatile' particles.*" (Page 11, L12-15)

page 13, line 26, and Fig. 8. I have some difficulties understanding how Fig. 8b was arrived at, and what should be compared in the figure. Which bars at at 10m and which ones at 220? The statement 'we assume that the volatility distribution (...) at 220 is a representative (...) for background particles' seems contradicting to fig 8a, where ca 33% of the total aerosol seems to still be traffic-originated. I think that it would be helpful to understanding if a more detailed (step-by-step) explanation for how each of the three mass concentrations were obtained could be given, maybe even in equation form in the supplementary. For example:
M(near-road)i = (SMPS total mass - BC - ACSM(inorganics) ) x (May et al)i
M(background)i = (bg OA mass ) x (TD vol. distribution at 220m)i

M(traffic)i = (NR - BG oa mass) x (TD vol. distribution at 10 m)i

We thank the reviewer for these helpful suggestions. In response to this and the other reviewer's comment, we have made multiple efforts to improve the clarity of the analysis and discussion around Fig. 8. For example, by changing the figure caption and legend, adding tags to specific bars, making changes in the text and adding a new supporting section in SI. The main point of this analysis is a comparison of our measured road-side volatility distribution (at 10 m) with a reconstructed distribution using traffic and background. We added multiple arrows in the revised figure that should clarify what should be compared in this figure (i.e., measured vs. reconstructed distribution). Following the reviewer's suggestion, we added a new section in the SI (sec. S4) that provides a detailed (step-by-step) explanation on how each of the three distributions were estimated.

The reviewer correctly pointed out that at our far-road location a non-negligible fraction (~20-30%) of total aerosol is likely from traffic emissions in this particular case. However, since we do not have measurements of particle volatility at our upwind background site, the use of the measured distribution from the far-road location as 'background' was our best approximation. We have added text to make this caveat clear.

*Since we did not measure particle volatility at our upwind background site, we assume that the volatility distribution measured at our far-from-road downwind location (220 m) is a representative distribution for background particles. This is a reasonable approximation as particle concentrations approach background levels within 200-300 m from the highway (Saha et al., 2017a). It should be noted that this 'background' OA contains a non-negligible (~25%) contribution from traffic emissions (Fig. 8a), and so likely has a slightly greater contribution from higher-volatility components than what would be measured in an actual 'background' location.*

One should highlight here that volatility of traffic-particles at near-road and the far-road location are not expected to be the same. The distribution at far from the road (more diluted) is expected to be more 'background like.' Therefore, even though a small fraction of the aerosol in the far-road location can be from traffic origin, its overall effects on the background volatility distribution is not expected to be dramatic as substantial evolution has already occurred. We have added a caveat that more extensive modeling is needed to represent these processes.

*While this simplified 'superposition' analysis suggests that our data are able to capture the influences of near-road evolution on emissions, further measurements and modelling work are needed to represent these dynamic processes.*

Revision: For further details, please see (i) revised Fig. 8, (ii) revised text in MS Sec. 3.4 and (iii) new addition of Sec.S4 in SI

Several assumed densities could be found in the article for different data analysis approaches (at least 1.1, 1.5 and 1.8 g/cm3 are described). Could this be made more consistent, or are there specific reasons for using these values that could be stated?

We assume a density of 1.8 g/cm3 for BC particles for SP2 data analysis. We estimated an effective density of 1.5 g/cm3 for submicron PM (see Fig. S7 for details) for converting SMPS total volume to mass.  A density of 1.1 g/cm3 (Table S1), was listed from May et al. (2013) along with different assumed TD kinetic model input parameters. However, since our TD kinetics modeling was based on tracking change in monodisperse particle size upon heating (V-TDMA approach), density does not play a role in the evolution of particle size. Therefore, we removed that entry from Table S1 to avoid confusion.

[revised manuscript text omitted]
_4$). This calculation assumes that aerosols are neutral. An ammonium balance can provide insights into the validity of assumption of neutral aerosol. Panel (b) shows $NH_4$ measured vs. $NH_4$ predicted plot. The $NH_4$ predicted is estimated as: $NH_4$ predicted = $2*(18/98)*SO4 + (18/63)*NO3 + (18/35)*Cl$, where the fractional amounts correspond to the molecular weights of the relevant species. An acidity plot (of measured vs. predicted $NH_4$) has an average slope

10 closer to ~1 (1.1±0.03), indicating an ammonium balance. Therefore, assumption of neutral aerosols at the measurement location was reasonable. SMPS mass concentrations in panel (a) were based on an estimated effective density of submicron aerosols of 1.5 cm$^{-3}$. The effective density is calculated by weighting fractional contribution of different species (e.g., campaign average: OA ~74%, AS ~13%, AN ~7%, and BC ~6%) with their respective densities. Assumed densities for AS, AN, and BC were 1.77, 1.72 and 1.8 g cm$^{-3}$, respectively. An effective density of OA of 1.45 g cm$^{-3}$, estimated from a parameterization by Kuwata et al. (2012) using elemental composition (O:C; H:C).

15 Kuwata et al. (2012) parameterization for OA density was developed based on laboratory data with negligible BC or NO3. It should be noted that Kuwata et al. applied their parameterization to data from the AMAZE campaign, which had an average OA fraction of 0.8 (versus 0.74 for our data), and found that the results agreed well with their measured density. Since our mass comparison based on application of this density to SMPS-measured volume and ACSM+BC mass showed good agreement (Fig. S7a), this indicates our overall estimated effective density, on which OA density has the largest influence, is well constrained.

[revised manuscript text omitted]

**S4: Reconstruction of road-side OA volatility distribution combining traffic and background volatility distribution.**

This section describes the analysis approach step-by step those are used for reconstruction of road-side OA volatility distribution combining traffic and background volatility distribution, as shown in Figure 8 and discussed in Sec. 3.4 (main paper)

**a)** Approximate OA mass concentrations at different distance from the highway are estimated from SMPS measurements as:

SMPS total mass conc. (function of distance) – BC (function of distance) – inorganics (ACSM at fixed location).

For example, Roadside OA mass at 10 m = SMPS total mass (10 m) - BC (10 m) - inorganics (ACSM at fixed site);

Background OA = SMPS total mass (background) - BC (background) - inorganics (ACSM at fixed site).

An effective density of 1.5 g cm$^{-3}$ (Fig. S7) was used to convert SMPS total volume to total mass.

**b)** Traffic OA at a downwind road-side location is estimated as: total road-side OA - upwind background OA

For example, traffic OA (at 10 m downwind) = total road-side OA (at 10 m downwind) – upwind background OA

**c)** Distribution of road-side OA, traffic OA, and background OA in different volatility bins (i) are calculated as:

$M_{OA}$ (road-side at 10 m)$_i$ = road-side OA mass at 10 m $\times$ (TD volatility distribution at 10 m)$_i$

$M_{OA}$ (background)$_i$ = background OA mass $\times$ (TD volatility distribution at 220 m)$_i$

$M_{OA}$ (traffic at 10 m)$_i$ = traffic OA mass at 10 m $\times$ (Traffic POA volatility distribution from May et. al.)$_i$

**d)** Reconstructed roadside OA volatility distribution is estimated as:

Reconstructed $M_{OA}$ (road-side at 10 m)$_{i\,=}$ $M_{OA}$ (traffic at 10 m)$_i$ + $M_{OA}$ (background)$_i$

Finally, we compare this reconstructed distribution with our road-side distribution measured at 10 mm (see Figure 8b of main paper).

**S5: Tracer m/z based factor analysis of ACSM data set**

Tracer m/z based OA components are estimated following Ng et al. (2011) as: hydrocarbon-like OA (HOA ~ $13.4 \times (C_{57}$ $- 0.1 \times C_{44})$) and oxygenated OA (OOA ~ $6.6 \times C_{44}$), where $C_{57}$ and $C_{44}$ are the equivalent mass concentration of tracer ion m/z 57 and 44, respectively. Previous evaluation of this method has shown that it can reproduce the HOA and OOA concentrations to within ~30% of the results from detailed PMF analysis at most sites (Ng. et al. 2011). The estimated traffic-OA (HOA factor) contribution was found to be substantially lower (~5-10x) than that derived based on background-subtracted roadside concentrations measured by SMPS in combination with other data (e..g, Fig. 8a in the main text). Therefore, in analysis shown in Figure S12, the volatility of near-road particles is dominated by contribution from background particles and the combined distribution does a poor job of recreating the observed near-road volatility distribution.

Several factors likely contribute to this discrepancy. For one, since the correlation equations for tracer m/z based factor analysis are empirical (Ng et al., 2011), this method's accuracy and representativeness may be limited in many environments. Further, a substantial fraction of traffic-emitted smaller particles may fall outside of transmission window of ACSM (< 70 nm), but these small particles likely do not have significant contributions to overall mass contribution. On the other hand, measurements of fresh vehicle-emitted particles with an SMPS may be biased high to some extent due to the non-spherical morphology of fresh soot particles (DeCarlo et al., 2004; Maricq and Xu, 2004; Park et al., 2003). For example, effective densities of vehicle-emitted particles with mobility diameters of 200 nm may be <0.3 g cm$^{-3}$, a factor of 5 lower than that value assumed here (Maricq and Xu, 2004). The true estimates of traffic-OA likely fall between the ACSM-HOA and SMPS-based estimation. However, these two estimates may be considered as bounding cases. Further efforts should be made to investigate the closure of estimates from different instruments and approaches.